# Integrated Morphological, Comparative Transcriptomic, and Metabolomic Analyses Reveal Mechanisms Underlying Seasonal Patterns of Variation in Spines of the Giant Spiny Frog (*Quasipaa spinosa*)

**DOI:** 10.3390/ijms25169128

**Published:** 2024-08-22

**Authors:** Gang Wan, Ze-Yuan Jiang, Nuo Shi, Yi-Ge Xiong, Rong-Quan Zheng

**Affiliations:** Provincial Key Laboratory of Wildlife Biotechnology and Conservation and Utilization, Zhejiang Normal University, Jinhua 321004, Chinamxjy3404@gmail.com (N.S.);

**Keywords:** *Quasipaa spinosa*, spines, morphology, comparative transcriptomics, metabolomics

## Abstract

*Quasipaa spinosa*, commonly known as the spiny frog, is an economically valued amphibian in China prized for its tender meat and nutritional value. This species exhibits marked sexual dimorphism, most notably the prominent spiny structures on males that are pivotal for mating success and species identification. The spines of *Q. spinosa* exhibit strong seasonal variation, changing significantly with the reproductive cycle, which typically spans from April to October. Sexually mature males develop densely packed, irregularly arranged round papillae with black spines on their chests during the breeding season, which may then reduce or disappear afterward, while females have smooth chest skin. Despite their ecological importance, the developmental mechanisms and biological functions of these spines have been inadequately explored. This study integrates morphological, transcriptomic, and metabolomic analyses to elucidate the mechanisms underlying the seasonal variation in spine characteristics of *Q. spinosa*. Our results demonstrate that spine density inversely correlates with body size and that spine development is accompanied by significant changes in epidermal thickness and keratinization during the breeding season. Comparative transcriptomic analysis across different breeding stages revealed significant gene expression alterations in pathways related to extracellular matrix interactions, tyrosine metabolism, Wnt signaling, and melanogenesis. Metabolomic analysis further identified significant seasonal shifts in metabolites essential for energy metabolism and melanin synthesis, including notable increases in citric acid and β-alanine. These molecular changes are consistent with the observed morphological adaptations, suggesting a complex regulatory mechanism supporting spine development and functionality. This study provides novel insights into the molecular basis of spine morphogenesis and its seasonal dynamics in *Q. spinosa*, contributing valuable information for the species’ conservation and aquaculture.

## 1. Introduction

The formation and evolution of sexual dimorphism are perennial subjects of interest among evolutionary biologists. Amphibians present a diverse range of sexually dimorphic phenotypes. For instance, some male frogs develop specialized protrusions, known as nuptial pads, on their thumbs during the breeding season. Additionally, in certain species, spiny clusters may also appear on other digits or the chest area. The genus *Quasipaa*, predominantly located in southern China and Southeast Asia, exhibits a high degree of polymorphism in male secondary sexual characteristics, such as spines distributed across the chest, abdomen, or the entire ventral surface [1]. Initially, studies posited that nuptial pads and thoracoabdominal spines primarily function to enhance friction and ensure secure mating grips [2]. Subsequent research, however, suggests that these structures may have more intricate roles, potentially involving chemical communication via secreted substances, possibly including hormones [3,4].

The *Q. spinosa*, commonly referred to as the spiny frog, is classified within the order Anura, family Dicroglossidae, and genus *Quasipaa*. This species is highly prized in China for its tender meat and substantial nutritional value, leading to a significant expansion in its artificial breeding, with an annual output now surpassing 6000 tons, underscoring its considerable market potential [5]. The reproductive cycle of *Q. spinosa* is highly seasonal. Typically, during the breeding season from April to October, males attract females with their calls and grasp them under the forelimbs with their strong forearms during mating. Sexually mature males have densely packed, irregularly arranged round papillae on their chests, with black spines that may reduce or disappear after the breeding season, while the chest skin of females remains smooth (Figure 1). Several studies have examined the histological structure of these keratin spines [6,7,8], noting that most sexually dimorphic features in amphibians manifest solely during the breeding season and are regulated by hormone levels [9,10]. Understanding these changes is crucial for elucidating the developmental mechanisms and biological functions of these sexually dimorphic traits.

Moreover, the spines of *Q. spinosa* are an important taxonomic characteristic within the tribe Paini [11]. Che et al. [12,13] conducted extensive research on the tribe, concluding that the genus *Quasipaa* is most closely related phylogenetically to the genus *Nanorana*, based on analyses of ancestral traits of male secondary sexual characteristics. They advocate for a classification that includes both genera *Quasipaa* and *Nanorana*, with the former distributed across southern China and Southeast Asia and the latter found in western China, the Qinling region, and the Tibetan Plateau. Typically, in *Quasipaa* species, male secondary sexual characteristics such as spines are located on the chest, abdomen, or encompass the entire ventral surface, whereas in *Nanorana*, they appear primarily in dual clusters on the chest. This distinction highlights the potential for further exploration into the molecular evolutionary mechanisms of sexual selection through the study of *Quasipaa* spines.

While previous studies have investigated the systematic evolution of *Q. spinosa*, the molecular mechanisms that govern spine formation have remained elusive. This study represents the inaugural effort to examine the biological functions of these spines through morphological analysis and to identify potential key metabolic pathways, metabolites, and genes involved in the formation of spines and their periodic changes using comparative transcriptomic and metabolomic approaches. Consequently, this research provides foundational data for the breeding and comprehensive application studies of *Q. spinosa*.

## 2. Results

### 2.1. Morphological Differences in Quasipaa spinosa Skin across Different Periods

Marked sexual dimorphism was observed in adult *Q. spinosa*, especially pronounced in males during the breeding season. Males displayed large, scattered spiny papillae on their chests, characterized by central black keratinous spines. These spines were more prominent and exhibited darker coloration compared to those observed during the non-breeding season. In stark contrast, female *Q. spinosa* displayed none of these features (Figure 1).

Microstructural analysis revealed that the spines consist of polygonal keratinized epidermal cells arranged in an inverted “V” configuration. Throughout the breeding season, the skin in the elevated spine regions of males showed a well-developed stratum corneum, granular layer, and stratum germinativum. The outermost layer was characterized by a deep-red keratin structure, displaying uniform and extensive keratinization. The thick stratum corneum comprised densely packed keratinized epithelial cells, lacking visible intercellular bridges. The cohesion between the stratum corneum and granular layer was evident, featuring tightly arranged granular and stratum germinativum layers. The innermost stratum germinativum cells appeared as short, columnar, and densely packed. The dermis was significantly thicker, enriched with abundant microvessels and blood vessels, and had pigment cells sporadically distributed in patches.

Post-breeding season observations indicated a reduction in the prominence of male spines, correlating with a smoother skin surface and a decrease in the number of granular and mucous glands (Figure 2).

### 2.2. Correlation Analysis between Spine Polymorphism and Body Morphometric Indices

A study was performed to explore the correlation between spine polymorphism and body morphometric indices in adult male *Q. spinosa*. The normality and variation of all measured data were evaluated using the Kolmogorov–Smirnov (K-S) test, which confirmed a normal distribution (*p* > 0.05) across the dataset [14].

An assessment of variability among nine morphometric indices and two spine morphological parameters ranked them from highest to lowest variability as follows: spine density (SD) > body mass (BM) > average spine elevation height (ASEH) > foot length (FL) > hand length (HL) > forearm circumference (CLA) > tibia length (TL) > tibia width (TW) > forearm plus hand length (FLAH) > snout–vent length (SVL) > hind limb length (LL). Notably, SD and ASEH exhibited significant variability, as detailed in Appendix A.

Further analysis using linear regression indicated a negative correlation between SD and several morphometric indices, including BM, CLA, HL, TW, and ASEH. This implies that the number of spines per unit area decreases as these body size parameters increase (Figure 3; Appendix A).

### 2.3. Analysis of the Skin Transcriptome in Quasipaa spinosa

#### 2.3.1. Transcriptome Data Quality Control

To investigate the differential gene expression related to seasonal spine variations, a total of 27 skin samples were collected from the chests of both sexes and the abdomens of males during the pre-breeding, breeding, and post-breeding periods. These samples were then subjected to transcriptomic sequencing. The quality of these biological replicates was initially evaluated using Pearson correlation coefficients, which demonstrated high correlations, as depicted in the dendrogram and scatter plot (Figure 4a,b).

The sequencing produced 157.81 Gb of clean data, with a minimum of 5.80 Gb per sample. The Q30 base percentage surpassed 91.64%, denoting high-quality sequencing (Appendix A). The assembly generated 117,123 transcripts with a total length of 162,346,008 bp, which were organized into 56,879 unigenes spanning 82,959,469 bp. Notably, 25,968 unigenes exceeded 1 kb in length (Figure 4c). Functional annotation of these unigenes resulted in 19,560 annotations (Figure 4d). The quality and depth of the RNA-seq data were considered sufficient for subsequent analyses, providing adequate variance for identifying differentially expressed genes (DEGs).

#### 2.3.2. Mining DEGs in Skin Transcriptomes

In comparisons across different stages and conditions (pre-breeding period female thoracic skin (AFC) vs. pre-breeding period male thoracic skin (AMC), breeding period female thoracic skin (BFC) vs. breeding period male thoracic skin (BMC), and late reproductive period female thoracic skin (CFC) vs. late reproductive period male thoracic skin (CMC)), significant numbers of DEGs were identified: 1674, 1829, and 1131, respectively, with more genes upregulated than downregulated in each scenario. Similar trends were noted in intra-male stage comparisons (pre-breeding period male abdominal skin (AMA) vs. breeding period male abdominal skin (BMA), breeding period male abdominal skin (BMA) vs. breeding period male thoracic skin (BMC), and late reproductive period male abdominal skin (CMA) vs. late reproductive period male thoracic skin (CMC)), with 825, 769, and 823 DEGs identified, respectively (Appendix A).

#### 2.3.3. Functional Categorization of DEGs in the Skin Transcriptome

DEGs were functionally categorized through KEGG pathway analysis to elucidate the mechanisms underlying sexual dimorphism in spine phenotype. Enrichment analysis revealed that upregulated DEGs during the pre-breeding stage predominantly enriched pathways associated with cell proliferation, including calcium signaling, ECM–receptor interaction, melanogenesis, and focal adhesion. During the breeding stage, the enriched pathways included adrenergic signaling in cardiomyocytes, Wnt signaling, and melanogenesis. In the post-breeding period, upregulated DEGs significantly enriched pathways such as Toll-like receptor signaling, HIF-1 signaling, and glycolysis/gluconeogenesis (Figure 5 and Figure 6; Appendix A).

#### 2.3.4. Analysis of Gene Expression Differences Related to Cell Proliferation and Keratin Synthesis among Comparison Groups

Marked variations in the expression of genes associated with keratinization were evident across different seasons. Key genes included those related to tendon proteins (*TNN* and *TNC*), collagen (*COL1A1*, *COL1A2*, and *COL6A3*), and integrins (*ITGA2* and *ITGB7*) during the pre-breeding stage. In the breeding season, genes involved in melanin synthesis (*TYR*, *TYRP1*, and *DCT*), muscle proteins (*MYH7B* and *MYL10*), and keratin (*KRT1*, *KRT12*, *KRT6A*, and *KRT5*) were upregulated. Following the breeding period, genes linked to autophagy (*MPO*, *C3*, *ATG10*, *CTSA*, *CTSL*, and *PRF1*) exhibited heightened expression (Appendix A).

### 2.4. Validation of DEGs through qPCR

Nine DEGs were selected randomly and validated via quantitative polymerase chain reaction (qPCR) to confirm the RNA-Seq data and the gene expression profiles illustrated in Figure 7. All nine sets of specific primers successfully amplified PCR products of the anticipated sizes, confirming their efficacy for DEG validation. The amplification efficiency of these DEGs was quantified as log2 fold changes, and when aligned with the RNA-Seq results, the expression patterns discerned by qPCR were consistent with those derived through RNA-Seq, thereby validating the accuracy of the RNA-Seq data. Consequently, these identified DEGs provide a crucial reference for future research into the molecular mechanisms underlying seasonal variations in *Q. spinosa* spines.

### 2.5. Comparative Metabolomic Analysis of Thoracic and Abdominal Skin in Quasipaa spinosa

#### 2.5.1. Identification of Differentially Expressed Metabolites (DEMs)

Metabolomic analysis was conducted on the thoracic and abdominal skin of males and the thoracic skin of females during the breeding season using LC-MS. Principal component analysis (PCA) revealed distinct separations among the three sample groups (Figure 8a). A total of 461 metabolites were identified as common across these groups, with specific DEMs identified in the BMA_vs_BMC and BFC_vs_BMC comparisons: 39 (22 upregulated and 17 downregulated) and 68 (63 upregulated and 5 downregulated), respectively (Figure 8b,c; Appendix A). Notably, 14 unique DEMs, including β-alanine, glucose-1-phosphate, citric acid, and d-glycerophosphate ester, were predominantly identified in the male chest skin during the breeding season, highlighting their potential roles in spine formation and maintenance (Figure 8d).

#### 2.5.2. DEM Functional Categorization through KEGG Enrichment Analysis

KEGG enrichment analysis was utilized to categorize the functional pathways of the identified DEMs from each sample group. DEMs in the BMA_vs_BMC comparison were significantly enriched in pathways related to steroid biosynthesis, oxytocin signaling, GnRH signaling, and phosphatidylinositol signaling system (Figure 9a). Conversely, DEMs in the BFC_vs_BMC comparison were predominantly involved in pathways such as the glucagon signaling pathway, citrate cycle (TCA cycle), central carbon metabolism in cancer, and pyruvate metabolism (Figure 9b). These pathways are implicated in various cellular functions, including proliferation, differentiation, energy metabolism, amino acid metabolism, antioxidant processes, and cell signaling.

### 2.6. Integrated Analysis of DEGs and DEMs

An integrated analysis was performed to examine the relationships between DEGs and DEMs identified in the BMA_vs_BMC and BFC_vs_BMC comparisons. Mapping these to the KEGG pathway database revealed 34 and 48 commonly enriched pathways, respectively (Figure 10a–c). Venn analysis highlighted 22 unique pathways in BMC compared to BMA and BFC, involving 46 DEGs and 6 DEMs related to energy metabolism pathways like citric acid and glucose-1-phosphate and cellular proliferation pathways such as α-tocopherol and d-glycerol 1-phosphate. Citric acid, notably, was associated with the largest number of target genes (Figure 10d), suggesting that gene expression may facilitate citric acid accumulation, thereby supporting cell proliferation, extracellular matrix synthesis, and other biosynthetic processes crucial for spine morphogenesis and maintenance. Pearson correlation analysis identified 13 genes with significant correlations (R^2^ > 0.9) to citric acid, with the highest correlation observed for gene c193625.graph_c3.

## 3. Discussion

### 3.1. Biological Functions and Morphological Characteristics of Spines

Amphibian skin displays a diverse array of cellular and extracellular structural components. Some species have developed unique skin adaptations, such as the keratinous spines in *Vibrissaphora boringii* [15] and the warty excrescences of *Andrias davidianus* [16]. These structures play essential roles in respiration, moisture retention, immune defense, and reproduction [17,18,19,20]. In male *Q. spinosa*, spines serve as secondary sexual characteristics closely linked to reproductive activities. Despite extensive research on sexual dimorphism in *Q. spinosa*, which has primarily focused on body size and breeding success [21,22,23], specific studies on the role of spines as secondary sexual characteristics are limited. Long [24] observed that males of *Q. spinosa* from Hunan’s Shimen region show significantly greater forearm and hand lengths, forearm widths, tibia lengths, and tibial widths compared to females. These features enhance predation capabilities and provide males with a competitive advantage in sexual selection, facilitating successful mating behaviors.

Yang [25] suggested that the primary cause of morphological differences between male and female *Q. spinosa* is differential strategies of energy allocation for reproduction, respiration, and circulatory organ development. These adaptations reflect profound differences in predation strategies between the sexes. Wang [21] observed significant morphological distinctions between the sexes, particularly in the more robust limbs of males, which provide advantages during amplexus in mating. Our study details the external morphology of male *Q. spinosa* across different periods, highlighting a significant increase in the keratinization and density of spines during the breeding season. Conversely, post-breeding season observations revealed a considerable reduction in gland and pigment cell densities in the spine region, alongside a thinning keratin layer and smoother skin surface, which may be reduced or disappear entirely. These modifications likely offer a competitive edge in mate competition.

Previous research has shown that nuptial pads on the forelimbs and spines on the ventral side are crucial in increasing friction and securing a firm grip during amplexus [10,26,27]. In this study, a correlation analysis between spine density and body size in male *Q. spinosa* during the breeding season revealed a significant negative linear relationship, suggesting that larger individuals typically have fewer spines. This observation indicates an ecological trade-off in resource allocation, where larger individuals may utilize physical strength or territorial advantages to enhance mating success, whereas smaller individuals may depend more intensely on a higher density of spines to improve amplexus efficiency and competitive viability.

### 3.2. Seasonal Variation in Spines Is Driven by a Complex Regulatory Network

In the skin, keratinocytes and fibroblasts synthesize essential proteins that help maintain the structural integrity and physiological functions of the dermis [28]. Collagen, predominant in the skin, muscles, and bones, serves as the most abundant component of the extracellular matrix. It not only forms various structural aggregates but also facilitates cell migration, differentiation, and proliferation, which are crucial for maintaining the normal structure and function of tissues and organs [29]. Experimental studies have demonstrated that collagen substrates can stimulate chondrogenesis and boost the expression of keratin, which is essential for spine development [30]. Our observations during the pre-breeding period showed upregulated expression of collagen-related genes, such as *COL1A1*, *COL1A2*, and *COL6A3*, in both male and female chest skin, likely contributing to the formation of spine structures and the structural remodeling of the skin.

Keratin, essential for spine development [31,32], exhibits significant differential expression in male *Q. spinosa* during the breeding season. Specifically, keratin genes (KRTs) such as *KRT1*, *KRT12*, *KRT6A*, and *KRT5* are upregulated in the chest skin of males compared to females. These genes play key roles in cytoskeletal remodeling and keratinization processes and are primarily expressed in epidermal keratinocytes. In clinical contexts, specific KRTs, including *KRT17* and *KRT6A*, are associated with keratinization disorders such as epidermolytic hyperkeratosis [33], ichthyosis [34], and pachyonychia congenita [35]. *KRT17*, a type I intermediate filament protein, is notably abundant in epithelial appendages like hair follicles and glands, contributing to cell proliferation and growth, skin inflammation, and differentiation [36,37]. During the breeding season, histological analysis of *Q. spinosa* skin demonstrated increased thickness in the stratum corneum, leading to spine-like structures, which is associated with the heightened expression of keratin genes. Moreover, the expression of melanogenesis-related genes (*TYR*, *TRP1*, and *DCT*) increases during this period, coinciding with the observed darker coloration and the distinctive black spine phenomenon in males. Consequently, the seasonal variation in the spines of *Q. spinosa* represents a complex biological event, coordinated by a network of multiple molecules and signaling pathways that collaborate to induce these morphological changes.

### 3.3. Joint Analysis of DEGs and DEMs Underscores the Role of Citric Acid Metabolism in Maintaining Spine Morphology

Our study utilized metabolomic techniques to examine differential metabolites in the skin of *Q. spinosa* across various parts and genders during the breeding season. The analysis emphasized the significant upregulation of metabolites involved in critical energy-related metabolic pathways, including glycolysis/gluconeogenesis, the citric acid cycle (TCA cycle), glyceride metabolism, and the metabolism of glycine, serine, and threonine. Among these, citric acid, a central intermediate of the TCA cycle, is crucial for cellular differentiation and proliferation [38], which are vital during the active phase of spine development. Beta-alanine, recognized for its role in enhancing physical performance, reacts with histidine to produce muscle-derived peptides such as carnosine, which enhances muscular endurance [39,40]. Glucose-1-phosphate, a key intermediate in carbohydrate metabolism, is transformed into glucose-6-phosphate, which then enters the glycolytic pathway [41]. This conversion highlights its importance in metabolic energy provision. Maleamate, a key component of NAD and NADP, plays a significant role in redox processes and cellular energy metabolism [42]. Creatine is identified as essential in facilitating ATP production through the respiratory chain [43].

Additionally, D-glycerol 1-phosphate, active in glycerophospholipid metabolism, aids in the formation of vital biomembrane structures and acts as an energy reservoir [44,45]. Its elevated levels during the breeding season indicate a heightened demand for cellular structure maintenance and energy storage to support increased metabolic activities. Alpha-tocopherol, an essential antioxidant, protects cellular membranes from oxidative stress, which is intensified during the breeding season due to elevated metabolic rates and environmental pressures [46].

The differential presence of these metabolites not only reflects adaptive changes in energy management and antioxidant protection but also signals physiological adjustments in *Q. spinosa* during the breeding season. Furthermore, the joint analysis of transcriptomic and metabolomic data demonstrates a significant accumulation of citric acid in the male thoracic epidermis during this period, corresponding with the upregulated expression of genes associated with structural proteins such as keratin and collagen. Considering the mechanical stress spines endure during mating and citric acid’s role in cellular energy provision, its accumulation is essential in driving the normal expression of structural proteins like keratin and collagen, thereby supporting spine morphogenesis and maintenance. This illustrates the intricate biochemical and molecular orchestration required to support the distinctive morphological adaptations observed in *Q. spinosa* during reproductive activities.

## 4. Materials and Methods

### 4.1. Sample Collection and Experimental Design

This investigation formed nine experimental groups, as detailed in Table 1. Specimens of *Q. spinosa* were collected from an artificial breeding facility in Jinhua, Zhejiang Province, China (latitude 28°53′ N, longitude 119°30′ E), during the pre-breeding (March), breeding (July), and post-breeding (November) seasons of 2022. Criteria for selection included only healthy, mature individuals with intact skin. Skin samples were obtained from the chest and abdominal areas of male *Q. spinosa* and from the chest area of females following euthanasia by double pithing. This ensured a minimum of three biological replicates per group [47]. Samples were divided: one segment was fixed in Bouin’s solution, which consists of 75% saturated picric acid, 25% formaldehyde, and 5% glacial acetic acid (Sigma-Aldrich, St. Louis, MO, USA), and the other was immediately frozen in liquid nitrogen (Air Products, Allentown, PA, USA) and then stored at −80 °C for further analyses. All experimental protocols adhered to the ethical regulations set forth by the Animal Welfare and Use Committee of the College of Life Sciences, Zhejiang Normal University.

### 4.2. Morphometric and Spine Parameters Measurement

During the breeding season (July), 81 healthy, 4-year-old mature male *Q. spinosa* were selected at random from the breeding facility. Following rapid anesthesia with 0.05% MS222 (Sigma-Aldrich, St. Louis, MO, USA), body weight, eight morphometric parameters, and two spine parameters were measured using an electronic scale (Mettler Toledo, Columbus, OH, USA) and digital calipers (Mitutoyo, Kawasaki, Kanagawa, Japan), respectively [48,49,50]. Spine density (SD) was defined as the number of spines per unit area of the spine region (spines/mm^2^ × 100%), and average spine height (ABH) was calculated by the total height of spines in a region divided by the number of spines (mm). A correlation analysis was conducted using SPSS 20.0 (IBM, Armonk, NY, USA) to determine statistical significance.

### 4.3. HE Staining and Histology

Skin samples as referenced in Section 2.1 were fixed in Bouin’s solution for 12 h and then embedded in paraffin. Sections (6–8 µm thick) were cut and stained with hematoxylin and eosin (HE) [51]. Microscopy was performed using an OLYMPUS BX51 microscope (Olympus Corporation, Tokyo, Japan), which was equipped with a cold light source digital camera (OLYMPUS DP70, Olympus Corporation, Tokyo, Japan).

### 4.4. RNA-Seq Library Construction and Sequencing

cDNA libraries were constructed from three skin samples from each of the nine experimental groups. RNA was extracted from tissue samples using the TRIzol reagent method (Thermo Fisher Scientific, Waltham, MA, USA). Tissue samples were homogenized in TRIzol reagent, followed by phase separation with chloroform (Thermo Fisher Scientific, Waltham, MA, USA). The aqueous phase containing RNA was then precipitated with isopropanol (Thermo Fisher Scientific, Waltham, MA, USA) and washed with ethanol (Sigma-Aldrich, St. Louis, MO, USA), and the RNA pellet was dissolved in RNase-free water (Thermo Fisher Scientific, Waltham, MA, USA). mRNA was isolated from total RNA using oligo(dT)-coated magnetic beads (Thermo Fisher Scientific, Waltham, MA, USA), which was then fragmented and purified. Following this, cDNA synthesis utilized random hexamer primers (Thermo Fisher Scientific, Waltham, MA, USA). The cDNA was then purified and underwent end-repair and adapter ligation, followed by selecting target size fragments via agarose gel electrophoresis (Bio-Rad, Hercules, CA, USA) and PCR amplification (Thermo Fisher Scientific, Waltham, MA, USA) to finalize the cDNA libraries. Sequencing was performed on an Illumina HiSeq 4000 platform (Illumina, San Diego, CA, USA) at BGI Tech Solutions (Guangzhou, China). The sequencing data were subjected to quality control, which included adapter removal and the filtering of low-quality reads. Transcriptome assembly was performed using the Trinity software suite (version 2.8.4, Broad Institute, Cambridge, MA, USA) [52], and the assembly quality was evaluated with BUSCO (Benchmarking Universal Single-Copy Orthologs) (version 4.1.4, Geneious, Auckland, New Zealand). Annotation of transcripts was conducted using multiple databases (NR, Swiss-Prot, Pfam, COG, GO, and KEGG) (Ensembl, version 101, Hinxton, UK). Transcript quantification was achieved using RSEM (version 1.3.3, University of Wisconsin-Madison, Madison, WI, USA) [53], with gene expression patterns visualized through Venn diagrams and Pearson correlation algorithms. Differential expression genes (DEGs) were identified using DESeq2 (version 1.26.0, European Molecular Biology Laboratory, Heidelberg, Germany), with criteria set at *p* < 0.05 and |log_2_FC| > 1 [54].

### 4.5. Validation of RNA-Seq Data by qRT-PCR

To confirm findings from the RNA-Seq analyses, nine DEGs were selected for quantitative analysis via qRT-PCR. Primers were designed using Primer 5 software (Premier Biosoft, Palo Alto, CA, USA), utilizing sequences obtained from the sequencing data, with *β-Actin* as the internal reference gene. Real-time quantitative PCR was conducted using the 2 × T5 Fast qRT-PCR Mix (SYBR Green I) kit (Vazyme Biotech Co., Ltd., Nanjing, China). The reaction protocol included an initial denaturation at 95 °C for 2 min, followed by 40 cycles of denaturation at 95 °C for 15 s, annealing at 60 °C for 15 s, and extension at 72 °C for 20 s, ending with a final melting stage from 65 °C to 95 °C. Each sample was analyzed in triplicate. Gene expression levels were determined using the 2^−ΔΔCt^ method, and results were reported as mean ± standard deviation (*n* = 3). Statistical significance was ascertained using one-way ANOVA (Table 1).

### 4.6. Metabolite Extraction and Analysis

Metabolite extraction and identification were conducted on skin tissues from three experimental groups (BMC, BFC, and BMA), each consisting of pooled samples from six randomly selected specimens, with each sample weighing approximately 50 mg. The samples were first homogenized in 1000 µL of extraction solvent (methanol/acetonitrile/water at a volume ratio of 2:2:1, containing an internal standard at a concentration of 20 mg/L). The homogenization process involved a ball mill at 45 Hz for 10 min, followed by sonication in an ice bath for 10 min and chilling at −20 °C for 60 min. After centrifugation at 4 °C and 13,400× *g* for 15 min, around 500 µL of the supernatant was collected, evaporated in an EP tube using a vacuum concentrator, and re-dissolved in 160 µL of extraction solvent (acetonitrile/water at a volume ratio of 1:1). The mixture was then vortexed for 30 s, sonicated in an ice bath for another 10 min, and centrifuged again at 4 °C and 13,400× *g* for 15 min. From this process, 120 µL of the clear supernatant was transferred to an LC/MS vial. For quality control, each sample contributed 10 µL to the analysis, which was performed using ultra-high-performance liquid chromatography–quadrupole time of flight–mass spectrometry (UHPLC-QTOF-MS).

The metabolites were separated using an Acquity UPLC I-Class PLUS system coupled with a Xevo G2-XS QTOF mass spectrometer (Waters, Milford, MA, USA) on an Acquity UPLC HSS T3 column (1.8 µm, 2.1 × 100 mm). The mobile phases included 0.1% formic acid in water (solvent A) and acetonitrile (solvent B), using a 1:1 (*v*/*v*) ratio. The elution gradient started at 2% B, increased to 98% over 10 min, was held for 3 min, then reduced to 2% B at 13 min, and finally stabilized at 5% B for the last 2 min. The column temperature and flow rate were maintained at 40 °C and 0.35 mL/min, respectively. The injection volume was set at 1 µL for both positive and negative ion modes, with capillary voltages at −2000 V for negative and 2500 V for positive ion modes. The collision energy varied from 2 V to 10–40 V, covering a mass range of 50–1200 *m*/*z*.

Peak extraction and alignment of raw data were performed using Progenesis QI software (Version 2.0, Waters Corporation, Milford, MA, USA). Reproducibility was assessed through Spearman correlation analysis and principal component analysis. Compounds were categorized, and pathway information was sourced from HMDB, KEGG, and LipidMaps databases. Orthogonal partial least squares–discriminant analysis (OPLS-DA) was utilized, using R language packages (Version 4.0.4, R Foundation for Statistical Computing, Vienna, Austria), to identify significant metabolites based on fold change (FC) > 1, *p* < 0.01, and variable importance in projection (VIP) > 1 [55].

### 4.7. Comprehensive Metabolome and Transcriptome Analysis

Principal component analysis (PCA) was employed to discern trends among the metabolites. Functional and signaling pathway enrichment analyses of differentially expressed genes (DEGs) were conducted using the KEGG database, with *p*-values less than 0.05 indicating significant DEGs. Pearson correlation coefficients were calculated between genes and metabolites to identify significant relationships. DEGs and differentially expressed metabolites (DEMs) with a correlation coefficient R^2^ > 0.8 were selected for further investigation. Canonical correlation analysis (CCA) was then employed to examine the molecular interactions between genes and metabolites during the formation and periodic changes of spines.

## Figures and Tables

**Figure 1 ijms-25-09128-f001:**
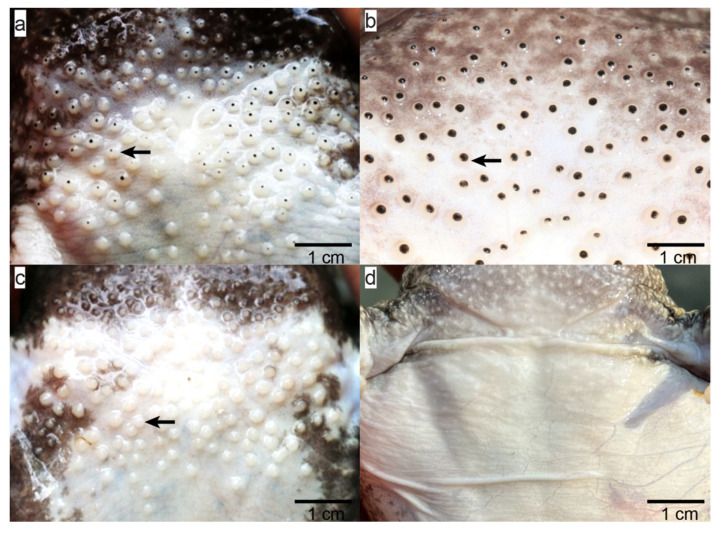
External morphology of the chest region of *Quasipaa spinosa* across different reproductive periods. Photographs (**a**–**c**) show the chest skin of males during pre-breeding, breeding, and post-breeding periods, respectively. Arrows indicate the spiny papillae on the male chest. (**d**) Chest skin of females, showing smooth skin without spines throughout the year.

**Figure 2 ijms-25-09128-f002:**
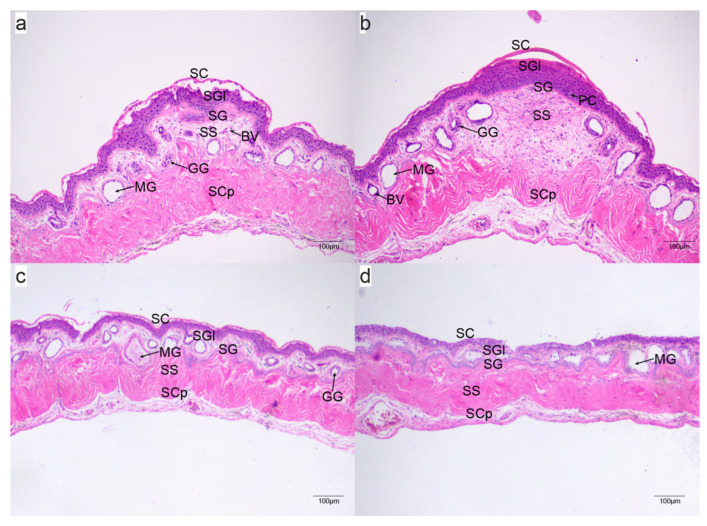
Microstructure of the chest skin of *Quasipaa spinosa* at different reproductive periods: (**a**–**c**) microstructure of the chest skin of males during the pre-breeding, breeding, and post-breeding periods, respectively; (**d**) microstructure of the chest skin of a female. SC: stratum corneum; SGl: stratum granulosum; SG: stratum germinativum; SS: stratum spongiosum; SCp: stratum compactum; MG: mucous gland; GG: granular gland; BV: blood vessel; PC: pigment cell.

**Figure 3 ijms-25-09128-f003:**
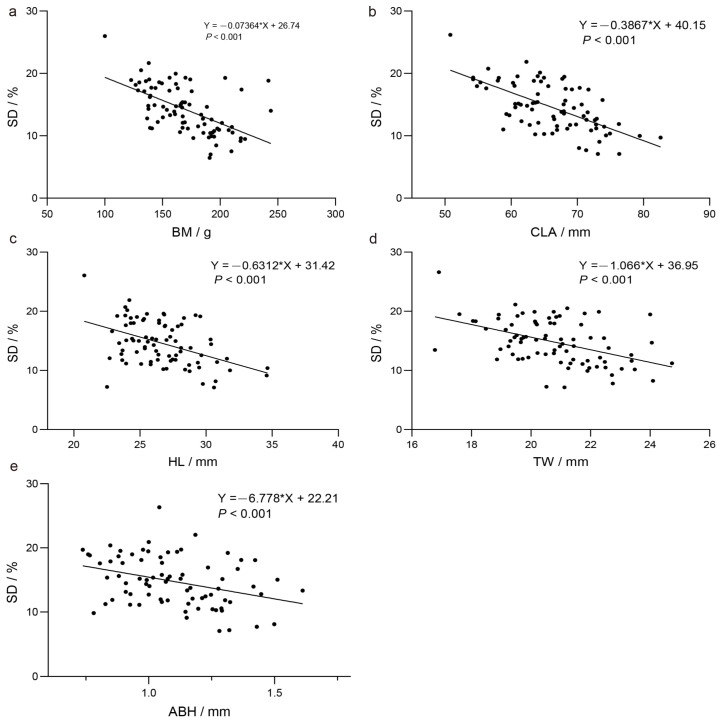
Pearson’s correlation analysis of spines with morphometric indices. Regression of body weight (**a**), forearm circumference (**b**), hand length (**c**), tibial width (**d**), and average height of spine projection (**e**) with spine density.

**Figure 4 ijms-25-09128-f004:**
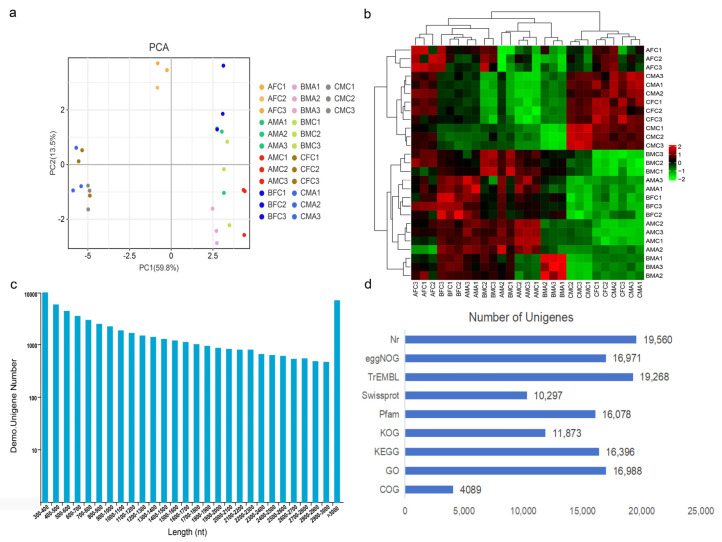
Unigene assembly and basic annotation. (**a**) PCA analysis demonstrating variations across 27 distinct sample sets. (**b**) Heatmap of correlation values (R^2^) for chest or abdominal skin samples. (**c**) Distribution of unigene lengths. (**d**) Statistics on the number of identified annotations.

**Figure 5 ijms-25-09128-f005:**
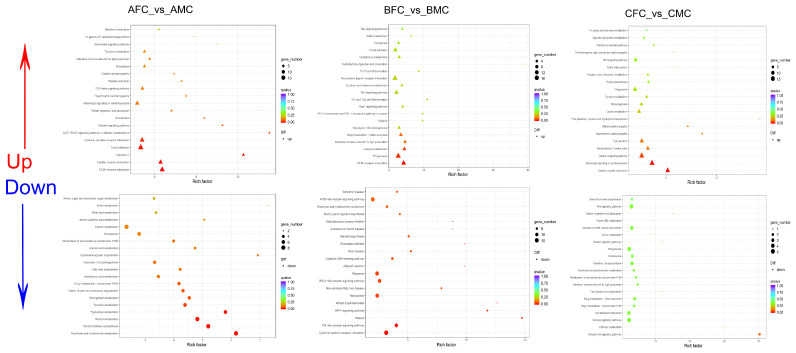
Top 20 KEGG pathways in a scatter plot. This figure highlights the enriched upregulated and downregulated DEGs across different genders during pre-breeding, breeding, and post-breeding periods in thoracic skin. The enrichment factor indicates the ratio of DEGs in a specific pathway to the total number of genes. The size of the points on these plots denotes the number of DEGs, while the color represents the q-value range.

**Figure 6 ijms-25-09128-f006:**
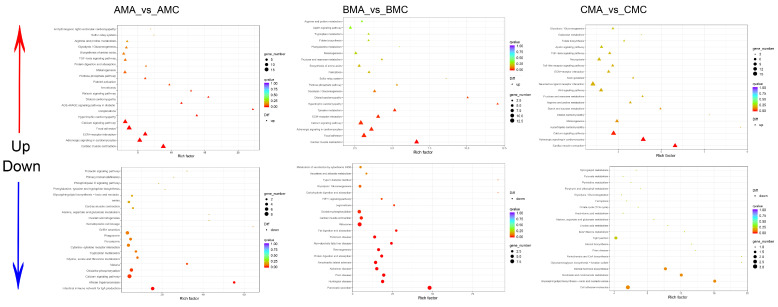
Comparative transcriptome analysis of different skin regions of male *Quasipaa spinosa*. The top 20 KEGG pathways enriched in upregulated and downregulated DEGs across various body parts of male *Q. spinosa* during pre-breeding, breeding, and post-breeding periods. The enrichment factor represents the ratio of the number of DEGs in a pathway to the total number of genes. The size of the dots indicates the quantity of DEGs, while the color reflects the q-value range.

**Figure 7 ijms-25-09128-f007:**
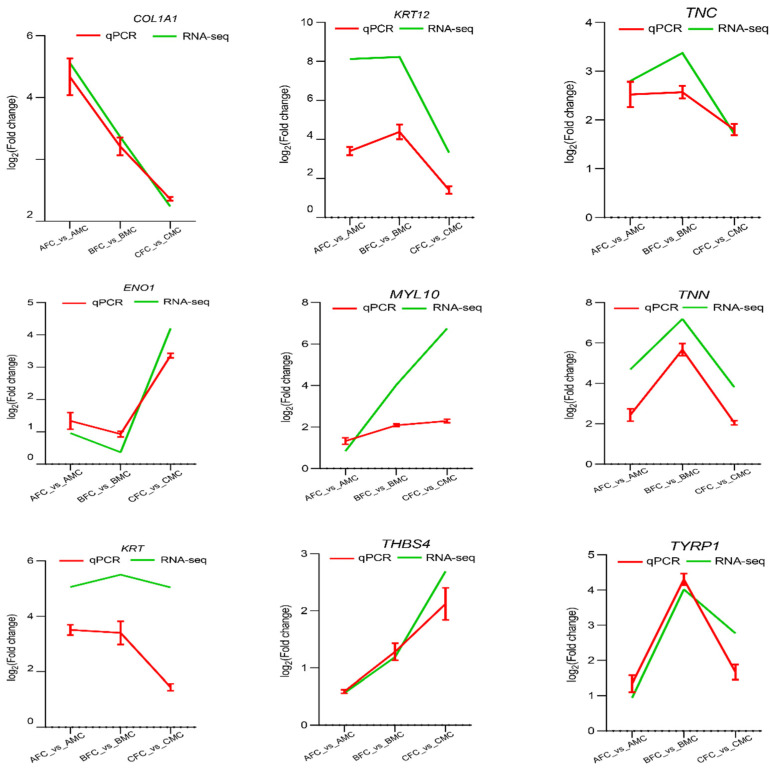
qPCR validation of RNA-Seq results. Nine DEGs were randomly selected for evaluation of their expression relative to the endogenous control gene (*β-Actin*) over various time periods. The relative expression values were converted to log2 (fold change) format. Results are presented as the mean ± SEM from the skin tissue of three *Q. spinosa*.

**Figure 8 ijms-25-09128-f008:**
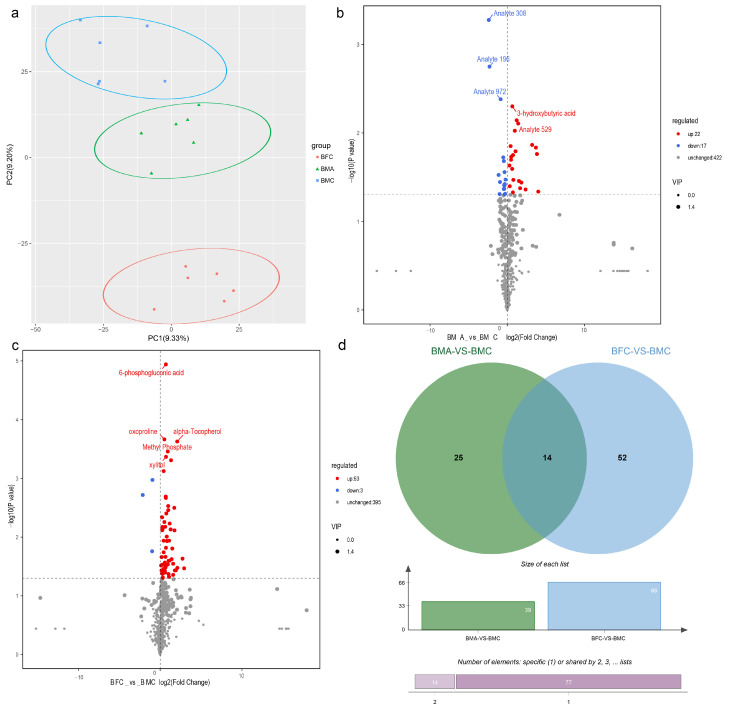
Metabolome analysis of BMA, BMC, and BFC. (**a**) PCA plot. Blue, green, and red points represent BMC, BMA, and BFC, respectively. (**b**) BMA_vs_BMC. (**c**) BFC_vs_BMC. (**d**) Venn diagram of DEMs.

**Figure 9 ijms-25-09128-f009:**
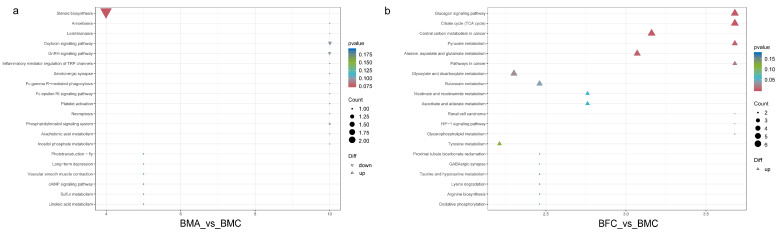
KEGG pathway enrichment analysis of DEMs for BMA_vs_BMC (**a**) and BFC_vs_BMC (**b**).

**Figure 10 ijms-25-09128-f010:**
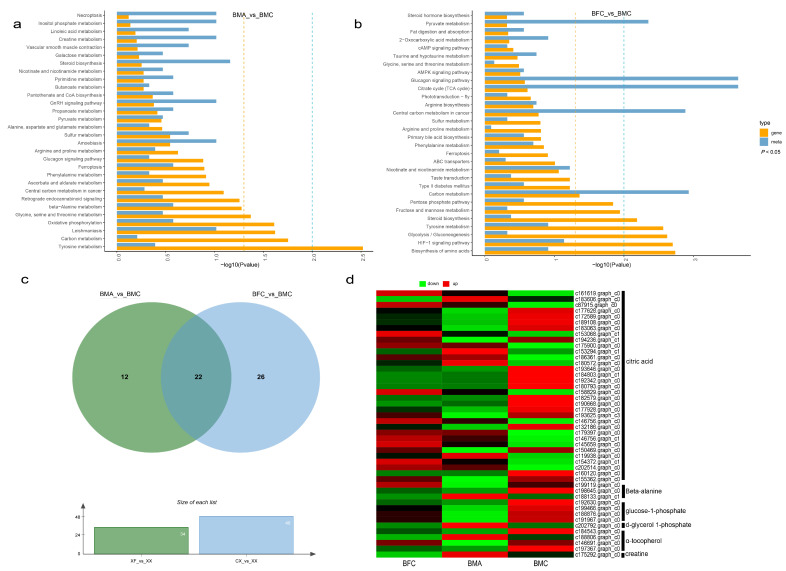
Integrated analysis of DEGs and DEMs for BMA_vs_BMC (**a**) and BFC_vs_BMC (**b**) with corresponding KEGG enrichment *p*-value histograms. “Meta” and “Gene” indicate the KEGG pathways enriched in DEMs and DEGs, respectively. The yellow line denotes significance (*p* < 0.05). (**c**) Venn diagram of joint KEGG analysis for BMA_vs_BMC and BFC_vs_BMC. (**d**) Heatmap displaying the expression levels of 22 genes (FPKM values). Red and green represent high and low expression, respectively.

**Table 1 ijms-25-09128-t001:** Information table of primers used in this study.

Primer Name	Primer Sequence (5′–3′)
*Qs*COL1A2-F	CTGGTTCATCTGGTGGTGGA
*Qs*COL1A2-R	TCGGGGTGGCTGAGTCTTA
*Qs*KRT12-F	CAAGAGAAGCAATGGGGAAG
*Qs*KRT12-R	AAACCAGAACAAGGAGGGATG
*Qs*TNC-F	GTTTCGCCGTTGTCTCTAAGG
*Qs*TNC-R	AACTGTCAAGGAGGTTGCTCTG
*Qs*ENO1-F	ACTCGGTCACGGAGCCAATCT
*Qs*ENO1-R	AGCCAGTGCAGGAATCCAGGTA
*Qs*MYL10-F	AAGCGGACAGATTCAGCCAAGA
*Qs*MYL10-R	CGCTACTCCTGGTCCTTCTCCT
*Qs*TNN-F	TTGGTCCAGTTCCTCCTCCTCT
*Qs*TNN-R	TTCTCCGTCTTCCGTGGGTTTG
*Qs*KRT-F	TTGCCCACTGTGCTTATTCC
*Qs*KRT-R	TTCTCTTCCATTCGCTTCTGAT
*Qs*THBS4-F	GCCCAAATCTACAACCACCC
*Qs*THBS4-R	CAATGAACCTCCCATGACCAC
*Qs*TYRP1-F	GGCAGAAGTACTAACCACAACG
*Qs*TYRP1-R	GAGAAGAGCCACATCTAAACGC
*Qs*β-Actin-F	CTGCTGAGCGTGAGATT
*Qs*β-Actin-R	GGCTGGAAGAGGGTTT

## Data Availability

The RNA sequencing data have been deposited in the NCBI SRA (http://www.ncbi.nlm.nih.gov/sra) under accession ID: PRJNA1150528. The other referenced data are included in the article or the Appendix A.

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
