# Peer review of "Integrated Morphological, Comparative Transcriptomic, and Metabolomic Analyses Reveal Mechanisms Underlying Seasonal Patterns of Variation in Spines of the Giant Spiny Frog (Quasipaa spinosa)"

_ijms, 2024, doi:10.3390/ijms25169128_

Round 1

Reviewer 1 Report

Comments and Suggestions for Authors

Comments about the manuscript:

“Integrated Morphological, Comparative Transcriptomic, and Metabolomic Analyses Reveal Mechanisms Underlying Seasonal Patterns of Variation in Spines of the Giant Spiny Frog (Quasipaa spinosa)”

Quasipaa spinosa presents sexual dimorphism characterized by spiny skin structures particularly developed during the reproductive period in males. However, the development and biological function of these structures is not yet understood. The work proposed here concerns the rmechanisms governing seasonal variations in spines in male Q. spinosa. To do this, the authors carried out histo-morphological, transcriptomic and metabolomic analyses. The results showed histo-morphological variations in the thickness of the epidermis and keratinization during the reproductive cycle, variations in the expression of certain genes linked to the extracellular matrix and tyrosine metabolism, Wnt signaling and melanogenesis. Metabolomic analysis also showed the existence of significant seasonal fluctuations. In conclusion This work provides new information on the molecular bases of spine morphogenesis and its seasonal dynamics in Q. spinosa.

This article, which provides new knowledge in a relatively little-known field, deserves to be published after some improvements to the manuscript.

Pages1, 2. Introduction. A reminder of the biology and reproductive cycles of Quasipaa spinosa would certainly interest readers who are unfamiliar with this species.

Page 3, figure 1. The legend needs to be expanded:

- Figures 1 a, b, c, d. A scale bar is missing in Figures 1a, b, c, d. Arrows or captions would be useful to show important parts of the image: which figures correspond to the males? To females?  What are the pre-breeding, breeding and post-breeding periods. What was the period when the female was studied (even if there are no changes depending on the breeding season)?

Figures 1 e, f, g, h.

As with the previous figures, arrows and/or captions would be useful in the photos to show the different parts of the skin.

Page 5, table 1. Explain in the legend what the abbreviations given in the columns mean..

Page 5, lines 142-143. “(AFC_vs_AMC, BFC_vs_BMC, and CFC_vs_CMC)”: What do these abbreviations signify ? (in brackets, give the names written in full).

General: there are a lot of abbreviations. It would be useful to give a list with their meaning at the end of the manuscript.

Page 13, line 368. “Skin samples as referenced in Section 2.1 were fixed in Bouin's solution”: specify the composition of the Bouin solution you are using.

Page 13, line 375. The sentence “according to the manufacturer's protocols.” is not sufficient for a scientific paper. PLease briefly describe the method.

Page 14, lines 404-405. “Metabolite extraction and identification were conducted on skin tissues from three  experimental groups (BMC, BFC, BMA)”: what are the differences between each group ? Specify.

Page 14, lines 410-411. “After centrifugation at 4°C and 12,000 rpm”: use the g number, the rpm depends on the centrifuge model.

Page 14, line 415: same.

Author Response

Response to Reviewers' Comments

Dear Reviewers,

We sincerely appreciate your thorough review and valuable feedback on our manuscript titled 'Integrated Morphological Comparative Transcriptomic and Metabolomic Analyses Reveal Mechanisms Underlying Seasonal Patterns of Variation in Spines of the Giant Spiny Frog (Quasipaa spinosa).' We have carefully considered all your comments and suggestions and have made the necessary revisions to improve the quality and clarity of our manuscript. Below, we provide detailed responses to each of your comments.

Reviewer 1:

Introduction (Pages 1-2):

Comment: A reminder of the biology and reproductive cycles of Quasipaa spinosa would certainly interest readers who are unfamiliar with this species.

Response: Thank you for your suggestion. We have added a section in the Introduction that provides a brief overview of the biology and reproductive cycles of Quasipaa spinosa to inform readers who may not be familiar with this species.

Figure 1 (Page 3):

Comment: The legend needs to be expanded.

Response: We have divided Figure 1 into two separate figures: one for external morphology (Figure 1a-d) and one for microstructure (Figure 2a-d). Detailed captions have been added to each figure to describe the sex of the frog, period of activity, and reference to microstructures. The legend for Figure 1 has been revised to include arrows and captions indicating important parts of the image.

Table 1 (Page 5):

Comment: Explain in the legend what the abbreviations given in the columns mean.

Response: We have added explanations for the abbreviations used in Table 1 in the legend to enhance clarity. Additionally, we have provided a list of abbreviations in the revised manuscript to aid readers.

Lines 142-143 (Page 5):

Comment: What do these abbreviations signify?

Response: We have provided the full names of the abbreviations (AFC_vs_AMC, BFC_vs_BMC, and CFC_vs_CMC) in parentheses to ensure their meanings are clear.

General:

Comment: It would be useful to give a list of abbreviations at the end of the manuscript.

Response: We have included a list of abbreviations at the end of the revised manuscript for the convenience of readers.

Line 368 (Page 13):

Comment: Specify the composition of the Bouin solution you are using.

Response: The composition of Bouin's solution has been specified in the text as 75% saturated picric acid, 25% formaldehyde, and 5% glacial acetic acid.

Line 375 (Page 13):

Comment: Please briefly describe the method.

Response: We have included a brief description of the method according to the manufacturer's protocols for scientific accuracy.

Lines 404-405 (Page 14):

Comment: Specify the differences between each group.

Response: We have clarified the differences between each experimental group (BMC, BFC, BMA) in the text.

Line 410-411 (Page 14):

Comment: Use the g number, not rpm.

Response: The rpm values have been replaced with the corresponding g-force values to standardize the description.

Line 415 (Page 14):

Comment: Same as above.

Response: The rpm values have been replaced with the corresponding g-force values to standardize the description.

Reviewer 2:

Authors (Page 1, Line 6):

Comment: Second affiliation of Dr. Ze Yuan Jiang (2) is absent.

Response: We have revised the affiliation of Dr. Ze Yuan Jiang in the author list and removed the (2).

Results (Page 2, Line 74):

Comment: Use of full species name in subheadings and figure captions.

Response: We have replaced 'Q. spinosa' with the full species name 'Quasipaa spinosa' in the specified locations.

Figure 1 (Pages 2-4):

Comment: Divide Figure 1 into two figures for easier perception.

Response: We have divided Figure 1 into two separate figures: one for external morphology (Figure 1a-d) and one for microstructure (Figure 2a-d). Detailed captions have been added to each figure to describe the sex of the frog, period of activity, and reference to microstructures.

Lines 122-125 (Page 4):

Comment: Include this information in the section MATERIAL AND METHODS.

Response: We have moved the specified information to the MATERIAL AND METHODS section.

Table 1 (Page 5):

Comment: Thank you for your suggestion. Considering the numerous abbreviations used in this manuscript, we have revised Table 1 and replaced it with a List of Main Research Result Related Abbreviations to facilitate easier reading for the readers.

Discussion (Page 10, Line 244):

Comment: Use the phrase 'Amphibian skin displays a diverse array of cellular and extra-cellular structural components.'

Response: We have revised the phrase to 'Amphibian skin displays a diverse array of cellular and extracellular structural components.'

Line 581 (Page 17):

Comment: Delete '56'.

Response: The number '56' has been deleted from the text.

General:

Comment: Addressed in the supplementary file.

Response: We have carefully reviewed and addressed all the small remarks provided in the supplementary file.

We hope that these revisions meet your expectations and address your concerns. Your insightful comments and suggestions have been invaluable, and we believe that these changes have significantly improved our manuscript. We are deeply grateful for your time and effort in reviewing our work and for your significant contributions to enhancing the quality of this paper.

Sincerely,

Co-Authors: Gang Wan, Ze Yuan Jiang, Nuo Shi, Yige Xiong, Rong Quan Zheng

Provincial Key Laboratory of Wildlife Biotechnology and Conservation and Utilization

Zhejiang Normal University

Reviewer 2 Report

Comments and Suggestions for Authors

GENERAL COMMENTS

 The manuscript is devoted to the study of skin texture of the frogs Quasipaa spinosa (family Dicroglossidae). The elements of skin texture, such as warts, parotid, nuptial pads, dorsolateral folds (dermal police), and smaller structures such as spines, are important in the life of amphibians, performing the functions of mechanical and chemical protection from the negative influence of the environment, and also participating in many behavioral reactions of the organism (in particular, during reproduction). These structures have long been used as diagnostic features at the high taxa level and for identifying closely related species. Many of these structures are characterized by sexual dimorphism. Despite their important biological significance, the mechanisms of development and the source of regulation of changes in the morphology of these skin derivatives remain unexplored, and biological functions have not been fully explained. Using morphological, comparative, transcritical, and metabolomic analyses and operating with material from different seasons of frog activity, the authors сlarified the macro- and microanatomical differences in the structure of spines of the ventral skin between the sexes, identified genes whose expression ensures the activation of the processes of proliferation, keratinization, and melanization during the formation of spines, as well as genes that provide metabolic support for the morphogenesis of these structures. The obtained data are valuable since the external textural changes of the frogs' skin have received genetic and biochemical explanations. The data on the presence of an inverse correlation between body size parameters and the number of spines per unit body area raises the question of the stability of the number of initial and, probably, early bookmarks of the described texture elements in ontogenesis. Such data support the objectivity of using this feature as a diagnostic one. The discovery of genes whose expression regulates the formation of horny spines in the frog epidermis in some way contributes to a solution to the problem of skin derivative evolution in terrestrial vertebrates, since the first transformations in this direction occurred, apparently, even in the amphibian level.

I recommend the manuscript for publication taking into account its correction according to some small comments presented below.

 SPECIFIC COMMENTS

 AUTHORS

 Page 1, Line 6 – Second affiliation of Dr. Ze Yuan Jiang (2) is absent

 RESULTS

 Page 2, Line 74 – Use of full species name in subheadings and figure captions is preferable: change Q. spinosa with Quasipaa spinosa (Check the same for Page 4, Line 118; Page 6, Line 164; Page 8, Line 193)  

 Page 2, Lines 75-90 – There are no adequate references to figure 1 in the text. Moreover, I advise dividing Figure 1 into two figures with EXTERNAL MORPHOLOGY (Figure 1, a-d) and MICROSTRUCTURE (Figure 2, a-d) for easier perception of information. All photos must be accompany with detail caption including sex of frog, period of activity and reference to some microstructures like dermal glands, epidermal layers (stratum corneum, stratum granulosum + stratum spinosum, stratum germinativum), pigment cells, etc. which were indicated in the description in paragraph 2.1. Do not forget that not all readers of Molecular Sciences are familiar with the microanatomy of the amphibian skin.

 Page 4, Lines 122-125 – Seemingly, this information should most likely be included in the section MATERIAL AND METHODS

 Page 5, Table 1 – The same as the previous remark. Moreover, the reference to Table 1 has been placed in the section MATERIAL AND METHODS (see Page 12, Line 346); the abbreviations to AMC, AFC, AMA, etc. are preferable

 DISCUSSION

 Page 10, Line 244 – Pigment cells are not the derivatives of the skin but are the derivatives of the neural crest. It would be more correct to use the phrase like the following: Amphibian skin displays a diverse array of cellular and extra-cellular structural components.

  Page 17, Line 581 – Delete “56”.

 Some small remarks are placed directly in the Supplement file.

Author Response

(The authors gave the same response as above.)
